# Differences in Stakeholders’ Perception of the Impact of COVID-19 on Clinical Care and Decision-Making

**DOI:** 10.3390/cancers14174317

**Published:** 2022-09-02

**Authors:** Joerg Haier, Johannes Beller, Kristina Adorjan, Stefan Bleich, Moritz de Greck, Frank Griesinger, Markus V. Heppt, René Hurlemann, Soeren Torge Mees, Alexandra Philipsen, Gernot Rohde, Georgia Schilling, Karolin Trautmann, Stephanie E. Combs, Siegfried Geyer, Juergen Schaefers

**Affiliations:** 1Comprehensive Cancer Center Hannover, Hannover Medical School, Carl-Neuberg-Str. 1, 30625 Hannover, Germany; 2Medical Sociology Unit, Hannover Medical School, 30625 Hannover, Germany; 3Department of Psychiatry and Psychotherapy, Ludwig-Maximilians-University Hospital, 80336 Munich, Germany; 4Department of Psychiatry and Psychotherapy, Hannover Medical School, 30625 Hannover, Germany; 5Department of Psychiatry, Psychosomatic Medicine and Psychotherapy, University Hospital, 60590 Frankfurt am Main, Germany; 6Department of Hematology and Oncology, Pius-Hospital Oldenburg, Carl von Ossietzky University, 26121 Oldenburg, Germany; 7Department of Dermatology, University Hospital Erlangen, 91054 Erlangen, Germany; 8Comprehensive Cancer Center Erlangen-EMN (CCC ER-EMN), 91054 Erlangen, Germany; 9Department of Psychiatry, Karl-Jaspers-Hospital, 26160 Oldenburg, Germany; 10Department of General, Visceral and Thoracic Surgery, Friedrichstadt General Hospital, 01067 Dresden, Germany; 11Department of Psychiatry and Psychotherapy, University Hospital, 53127 Bonn, Germany; 12Department of Respiratory Medicine and Allergology, University Hospital, 60590 Frankfurt am Main, Germany; 13Department of Hematology, Oncology, Palliative Care and Rheumatology, Asklepios Tumorzentrum, 22763 Hamburg, Germany; 14Department of Hematology and Oncology, University Hospital, 01307 Dresden, Germany; 15Department of Radiation Oncology, Technical University of Munich (TUM), Klinikum Rechts der Isar, 81675 Munich, Germany

**Keywords:** decision conflicts, moral distress, uncertainty, oncology, psychiatry, COVID-19

## Abstract

**Simple Summary:**

Pandemics induce many changes in clinical management. The consequences and extent of these changes are perceived in an individual manner and differ between various stakeholder groups. Using a cross-sectional questionnaire in 11 German institutions we evaluated the different perceptions of related risks and decision-making processes. All the investigated groups share concerns about the impact of the COVID-19 pandemic on healthcare management and clinical processes, but to very different extent. Their perception is dissociated in projection towards other stakeholders. Specific awareness should avoid this dissociation that potentially results in impaired shared decision-making.

**Abstract:**

Background: Pandemics are related to changes in clinical management. Factors that are associated with individual perceptions of related risks and decision-making processes focused on prevention and vaccination, but perceptions of other healthcare consequences are less investigated. Different perceptions of patients, nurses, and physicians on consequences regarding clinical management, decisional criteria, and burden were compared. Study Design: Cross-sectional OnCoVID questionnaire studies. Methods: Data that involved 1231 patients, physicians, and nurses from 11 German institutions that were actively involved in clinical treatment or decision-making in oncology or psychiatry were collected. Multivariate statistical approaches were used to analyze the stakeholder comparisons. Results: A total of 29.2% of professionals reported extensive changes in workload. Professionals in psychiatry returned severe impact of pandemic on all major aspects of their clinical care, but less changes were reported in oncology (*p* < 0.001). Both patient groups reported much lower recognition of treatment modifications and consequences for their own care. Decisional and pandemic burden was intensively attributed from professionals towards patients, but less in the opposite direction. Conclusions: All of the groups share concerns about the impact of the COVID-19 pandemic on healthcare management and clinical processes, but to very different extent. The perception of changes is dissociated in projection towards other stakeholders. Specific awareness should avoid the dissociated impact perception between patients and professionals potentially resulting in impaired shared decision-making.

## 1. Introduction

The COVID-19 pandemic has been connected with numerous changes in treatment of many patient groups, such as for cancer [1] and psychiatry patients [2]. These pandemic effects on the availability and accessibility of healthcare appears to very intensively induce challenges in shared-decision-making (SDM) [3,4], moral distress for healthcare providers, and decisional problems for patients [5]. Healthcare professionals were confronted with complex decisions and ethical dilemmas about the provision of care and their implicit judgements about access for patients [6]. For patients, decisional conflicts were reported, such as when undergoing surgery [7].

This uncertainty and moral distress is inherent in sudden healthcare crises [8], but may not be solely determined by objective pandemic indicators. It seems to be intensively influenced by individual perceptions of the pandemic situation and the related consequences [5,9]. The personal reflection of the pandemic consequences appears to be of high relevance since the individual perception of risk guides response and health-related decision-making [10,11]. The evaluation of COVID-19-related risks and decision-making processes mainly focused on prevention and vaccination behaviors [12,13]. However, in other clinical entities, compliance with treatment and the handling of evidence deficiencies also rapidly affected clinical management, individual behavior, and outcome [14].

Previously published results from our group [5] suggested that the clinical setting appears to influence the extent of decisional conflicts and impact on SDM. Distinct perception profiles of changes in oncology care processes due to COVID-19 were previously identified and more than 20% of the healthcare professionals, but only ~11% of the patients reported severe decisional conflicts during the pandemic [5]. Therefore, it seems to be interesting to compare the perception of decisional conflicts and the impact on SDM between different stakeholder groups.

Different perspectives for decision-making, context, and outcomes were mainly neglected in evidence-based management during the pandemic and the role of individual perceptions have not been investigated systematically yet. Pandemic modelling approaches should include all of the relevant perspectives and behavioral patterns to implement early qualitative awareness and preparedness [15]. In addition, alignment of perspectives and the perception of pandemic consequences from all involved groups may avoid confusion, loss of trust, and frustration, especially for SDM under pandemic conditions.

A prerequisite of SDM is an understanding between patients and healthcare professionals regarding clinical aspects as well as sharing similar perspectives on values and criteria to make these decisions. Therefore, we compared the different perceptions of patients, nurses, and physicians regarding consequences of the COVID-19 pandemic for clinical management, decisional criteria, and decisional burden.

For cancer care we hypothesized that the high prognostic, potentially life-threatening impact of diagnostic and treatment delays might be a determinant of the perception of decisional conflicts and SDM impact. In contrast, entities that are intensively related to perception disturbances can also have intensive effects on decision-making for all of the involved groups. Therefore, cancer care was compared with psychiatry to differentiate these two aspects.

## 2. Methods

### 2.1. Questionnaire

Questionnaire data were used to obtain perspectives of the different stakeholder groups regarding pandemic-related decisional uncertainties and the impact of COVID-19 on clinical care as previously described [5]. Briefly, the qualitative results were aggregated into dimensions covering conflicts/uncertainty, resources, risk perception, perception of consequences for clinical processes, and in clinical care with 3–5 questions for each dimension. If applicable, identical questions were used for the different stakeholder groups. In the evaluation of impact perception, overall 118 variables were included. Validation of the questionnaire versions (available only in German language) were done in two rounds with 5 representatives in each group.

### 2.2. Participants

Cross-sectional data were obtained from the OnCoVID trial (ethical approval 9199_BO_K_2020) and collected as a pen and paper survey between 10/2020 and 06/2021 from 1231 patients, physicians, and nurses in 11 participating hospitals. Recruitment was done in university and non-academic institutions (hospitals, out-patient centers) throughout Germany by mail-. The participants had to be involved directly in clinical care in oncology or psychiatry (without mentally impaired patients). Details of recruitment were previously published [5].

### 2.3. Variables

Ordinal variables were coded as 5-point scales according to the related questions (from “not at all” to “completely”; “not at all/seldom” to “most of the time”; “much less” over “no changes” to “much more”; “not likely” to “very likely” “very negatively” over “no changes” to “very positively”). Demographic data included gender (male, female); specialty (psychiatry, oncology); age (years); stage of treatment (“Initial treatment after diagnosis”, “Treatment continuation”, “Recurrence/metastasis/crisis treatment”, “Follow up”); and educational background (7 categories). Dichotomic variables were used as “yes” or “no”. All the scales were applied as equidistant [16]. Missing data occurred in up to 3% of the participants depending on the different items. These data were excluded from the analysis in a case-based manner.

### 2.4. Data Analysis

Parameters for decisional uncertainty, distress, and reflection of the participants’ psychological environment were described using histograms, boxplots, means ± SD, and 95% confidence intervals.

For post hoc tests, ANOVA and Tukey-HSD were used to compare participant groups and ordinal variables. Pearson rank correlation for comparison of two groups and *t*-tests for continuous variables were applied. In the case of categorical variables Chi²-tests with continuity correction were performed.

For items with high similarity, multivariate factorial analysis was performed as principle component analysis (PCA) in order to reduce the number of factors for further analysis. In a stepwise approach and based on a sufficient number of significant correlations approved by Kaiser–Meyer–Olkin criterion (KMO accepted if >0.5) and significance of Bartlett test for sphericity aggregated factors were extracted.

For multivariate comparison of stakeholder groups, discrimination analysis was applied. The variables that were identified in the univariate approaches were subsequently used for differentiation between the various stakeholder groups. Non-respondents were excluded pairwise from analyses of the respective items. Univariate ANOVA and Eigenwert provided information about the quality of the discrimination functions.

All evaluations were done using SPSS26.

## 3. Results

### 3.1. Questionnaire Response and Cohort Description

A total N = 1231 (730 female, 473 male, 28 N/A) representing a response rate of 54.4% (oncology 54.1%; psychiatry 55.0%) were included (540 patients (response 54.4%), 322 physicians and 369 nurses). There were 834 participants that were related to oncology and 397 to psychiatry. The age distribution was arranged in three different groups (≤40 years: N = 435; 41–65 years: N = 610; ≥66 years: N = 164).

### 3.2. Perception of Workload and Clinical Management by Professionals

All professionals were asked whether the individual workload during the pandemic has changed and the majority of them in all groups answered “A little more”. However, a large subgroup of 29.2% reported extensive changes in the workload with highest perception by psychiatry nurses (37.9%) and lowest by physicians in oncology (20.3%). Generally, nurses suffered from significantly more intensive changes in workload than physicians (*p* < 0.001) (Figure 1A).

Physicians reported a slight improvement of hygiene within the hospitals and worsening of trial conduction whereas nurses did not reflect such positive effects. Although all groups saw impaired general clinical management, other aspects, such as quality of care, data protection, informed consent management, keeping distance regulations, and multiprofessional exchange, were not involved in the perception of relevant pandemic-induced changes (Figure 1B).

### 3.3. Perception of Modified Clinical Care and Consequences

All professional groups (physicians and nurses in oncology and psychiatry) reported perception of changed clinical processes and resources for their clinical care due to the pandemic. However, the extent of this individual assessment was less than expected, and for most questions the mean values were “No changes” to “A little bit worse” (Figure 2A). Although in few entity-specific areas (surgery, psychotherapy, availability of beds in psychiatry) worse processes and resources were reported and the overall reflection was similar in all professional groups.

In their own field of expertise, nurses reported relatively low impact on their daily nursing care, whereas the impact on the patients’ psychosocial environment, such as for giving advice, providing psychosocial support, or assisting relatives, was considered moderately affected (Figure 2B). This was highly comparable to the projection of these consequences by nurses towards the patients (correlation R^2^ = 0.42–0.54). The evaluation of nursing requirements (*p* = 0.002), giving advice to patients (*p* = 0.003), creating relationships to patients (*p* < 0.001), and assisting in psychosocial issues (*p* = 0.028) were significantly worse, reflected by nurses in oncology compared to psychiatry. Similarly, the oncology nurses have seen worse consequences for patients than psychiatry nursing staff (obtaining nursing advice: *p* = 0.003; getting relationship to nurses: *p* < 0.001; care for relatives: *p* = 0.017).

Healthcare professionals in psychiatry returned a severe impact of the pandemic on all major aspects of their clinical care, which was similar in physicians and nurses. In contrast, these changes were reported mainly below “Somewhat” by professionals that were involved in oncological care. Nurses reported worse values in half of the categories (Figure 2C,D).

For comparison of the perception of modified clinical care due to the pandemic situation, patients were asked similar questions that were related to key aspects of their clinical management. Surprisingly, patients in in both entities reported much lower recognition of the generally required treatment modifications and consequences for their own clinical care. Although the questions were not fully identical and a formal statistical comparison cannot been done, the differences between professionals and patients can easily be seen in Figure 3A,B. Patients neglected the consequences despite some delays that occurred in their treatment. The vast majority of oncology patients (<80%) reported no changes except for rehabilitation (62.9%). Patients in psychiatry acknowledged more frequently changes in various treatment modalities, but more than 90% of them received their treatment with a maximum of 2 weeks delay (Figure 3C,D).

### 3.4. Decisional Conflicts and Burden

It was further investigated whether the different perception of pandemic-related changes in clinical infrastructure and processes affected the decision-making of the various stakeholders. We found a similar picture compared to the reflection of the clinical care. Patients reported only slight changes of the decisional criteria for treatment. The additional risks to obtain SARS infections and additional side effects/complications played only minor roles for their decisions in oncology and psychiatry. In contrast, physicians perceived these decisional changes to a significantly larger extent. (*p* < 0.001; Appendix A) Symptom control as decisional criterion was significantly different in oncology (*p* = 0.001), but not in psychiatry (*p* = 0.079) (Figure 4A).

Previously, we reported that decisional uncertainty and conflicts resulted in a decisional burden in patients and in healthcare professionals. Therefore, we asked the participants regarding their perception of different aspects of such burden that they have seen in other stakeholder groups. Overall, patients again reflected the burden of physicians (2.16 ± 1.53 points at 5-point scale) and nurses (2.18 ± 1.70) to a lower extent than professionals the burden of patients (2.78 ± 1.12). The highest burden was attributed from professionals to other professional groups (3.26 ± 1.29). Interestingly, the burden by infection risk for the opposite group was also less reported by patients (2.20 ± 1.44) than by professionals (2.83 ± 1.32). Since the questions were slightly different, a statistical group comparison was not done (Figure 4B).

If patients were asked regarding the decisional uncertainty and conflicts of healthcare professionals only 4.2–14.5% of the oncology patients observed higher values (“A lot” or “Completely”) whereas significantly worse values (*p* < 0.001) were found in psychiatry patients in all categories (Figure 4C,D).

### 3.5. Discrimination Analysis of Impact Perception

To differentiate the factors that may determine the perception of pandemic challenges, discrimination analyses were done. Targeting differences between patients in oncology and psychiatry, a discrimination function was found that was able to correctly classify 85.5% of the patients. The obtained function was highly predictive (Eigenwert = 0.907, Wilks–Lambda = 0.508, *p* < 0.001) and was mainly determined by the variables age group, current psychological conditions (aggregating anxiety, depression, stress, loneliness, and hope), and the burden by SARS infection risk (Table 1A). In a similar function for healthcare professionals, we compared nurses and physicians regarding their perception of the pandemic situation. After eliminating the variables that were not significantly different between both groups, 10 items remained to build up a discrimination function (Table 1B). The obtained function had less discriminative power and classified 71.9% of the professionals correctly (Eigenwert = 0.291, Wilks–Lambda = 0.774, *p* < 0.001). The most important factor was the perception of patients’ burden by the pandemic followed by the pandemic workload and various items that were related to the perception of available resources and management processes (changes in drug treatment, hygiene, availability of drugs, and diagnostics). Since the questions for patients and professionals were slightly different (although targeting the same domains), a direct multivariate comparison between these groups was not applicable.

## 4. Discussion

The perception of the pandemic situation within healthcare environments appears to be characterized by very individual perspectives. In our investigation we compared three major groups (patients, physicians, nurses) that participated in the same healthcare provision processes within the same clinical environments. However, their perception of pandemic consequences differed significantly between the stakeholder groups. In general, patients observed less problems in healthcare processes, resources, and quality of care than both groups of healthcare professionals. Nurses reflected, on average, the highest values of worsening conditions. This pattern of perception (patients < physicians < nurses) was seen in both entities, but to a different extent and more intensively in psychiatry than in oncology. This resulted in a projection of intensive consequences from nurses towards patients, whereas the patients themselves recognized these consequences to a much lower extent.

The specific vulnerability for perception of disturbed conditions during a healthcare crisis by nurses has been confirmed by other investigations [17]. Such dissociation between the perception of own risks during the pandemic and the risk of others was similarly reported for the general population [10] and for healthcare managers [18]. Similarly, a dissociation between individual pandemic risk perception and local indicators of COVID-19 risks was found [5,12].

Besides the individual observation of pandemic consequences for clinical care, we also found major differences in the perception of decisional uncertainty and conflicts in projection towards the opposite groups. One out of eight oncology patients and one out of four psychiatry patients attributed severe decisional dilemmas to their healthcare professionals; in a similar manner to physicians and nurses. Comparable to the recognition of healthcare deficits during the pandemic, the professionals’ projection towards the decisional burden for patients was more intensive than in the opposite direction by patients towards professionals. Patients appear to be more concerned about the economic, psychological, and interpersonal consequences of the COVID-19 pandemic, rather than about their own health [19].

Interestingly, younger age (<40 years) but not gender was identified as a predictor for decisional conflicts in all the participating groups. Unfortunately, the structure of the questionnaire processes did not allow a differentiation between different provider types. This should be considered for future research.

The dissociation of patients’ and professionals’ perception of impaired pandemic preparedness and their concerns about its impact on healthcare appears to be of high importance. It likely affects individual decisional behavior of professionals that can result in reduced adherence to clinical guidelines [20]. During a crisis, sufficient SDM that is based on stakeholders’ perspectives of pandemic impact seems to be important as negotiation between societal responsibility, perceived infection risks, and individual decisional burden [21]. The societal conflicts appears to be more vulnerable in nurses and should be addressed early as part of the pandemic preparedness. For patients, the relevance of trust in clinical care structures, confidence with handling of COVID-19 by healthcare professionals, and relatively low impact on their medical decision-making need to be considered in this adaptation process [22]. The alignment of hazard-related preparedness (based on objective criteria) and stakeholders’ individual perceptions (including resulting decisional dilemma) may act as a mediator for guiding this healthcare management adaptation during a pandemic [23].

In summary, patients, nurses, and physicians share concerns about the impact of the COVID-19 pandemic on healthcare management and clinical processes, but to a very different extent. This perception appears to be dissociated in projection towards the other stakeholders, and healthcare professionals seem to overestimate the impact of the pandemic on patients, their SDM, and related conflicts. Proper healthcare management support can avoid therapists’ pandemic frustration, maintain physical and mental health, and a healthy psychosocial work environment [24,25]. To avoid the dissociated perception of the pandemic impact between patients and professionals, potentially resulting in impaired SDM, a specific awareness should be provided and trained for professionals that are dealing with clinical care under the conditions of a pandemic [26]. Further research needs to be done for the evaluation of perception dynamics over the course of a pandemic and its relationship to incidence developments.

## 5. Conclusions

Patients and healthcare professionals reflect decisional conflicts that appear as a result of the pandemic and related changes in clinical process management. These decisional conflicts were seen in the projection of the own group. However, this perception towards the other stakeholder groups was dissociated regarding the severity of the impact and overestimated the effects of the pandemic. This dissociation can affect SDM, and adapted pandemic management seems to require specific support to maintain awareness for these decisional conflicts.

## Figures and Tables

**Figure 1 cancers-14-04317-f001:**
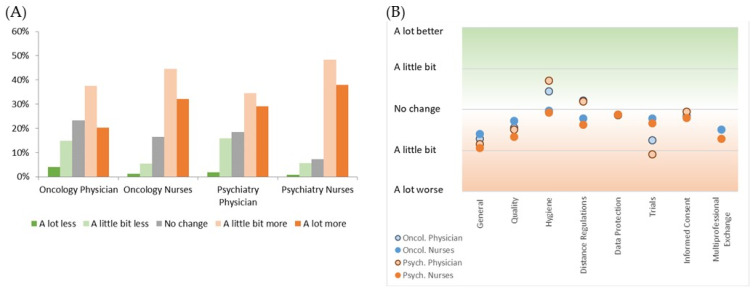
(**A**) Reflected alteration of the workload by healthcare professionals; and (**B**) their perception of changes in clinical management due to pandemic conditions.

**Figure 2 cancers-14-04317-f002:**
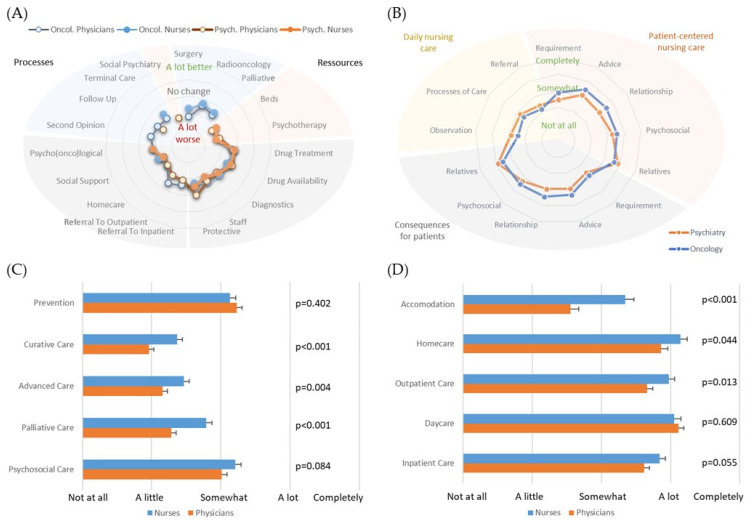
(**A**) Perception of altered processes and resources for clinical care by professionals (blue background: oncology, orange: psychiatry, grey: both entities); (**B**) Perception of effects in nursing care by nurses categorized as patient-centered nursing care (yellow), daily nursing care (orange), and consequences for patients (grey); Perception of changes in healthcare processes due to the pandemic in their own specialty by healthcare professionals in (**C**) oncology and (**D**) psychiatry. Significant differences were found between nurses and physicians.

**Figure 3 cancers-14-04317-f003:**
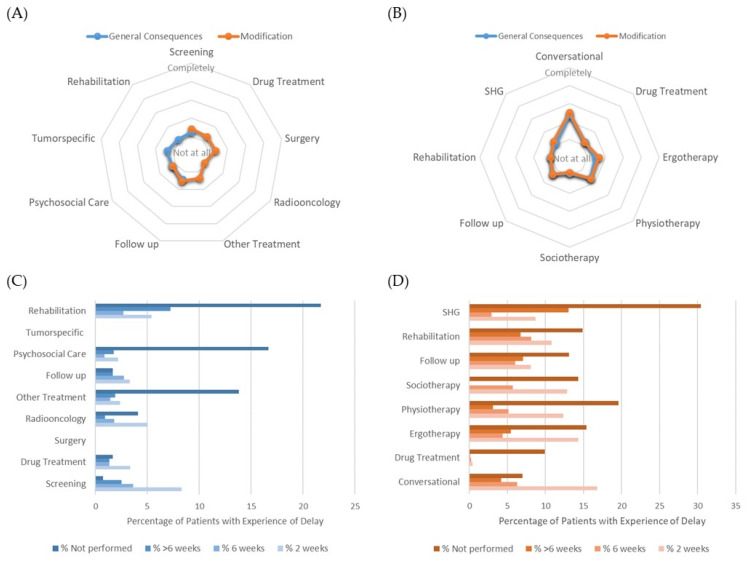
Perception of generally required treatment modifications and consequences for their own clinical care by patients in (**A**) oncology and (**B**) psychiatry; patients’ acknowledgement of treatment delay in (**C**) oncology and (**D**) psychiatry.

**Figure 4 cancers-14-04317-f004:**
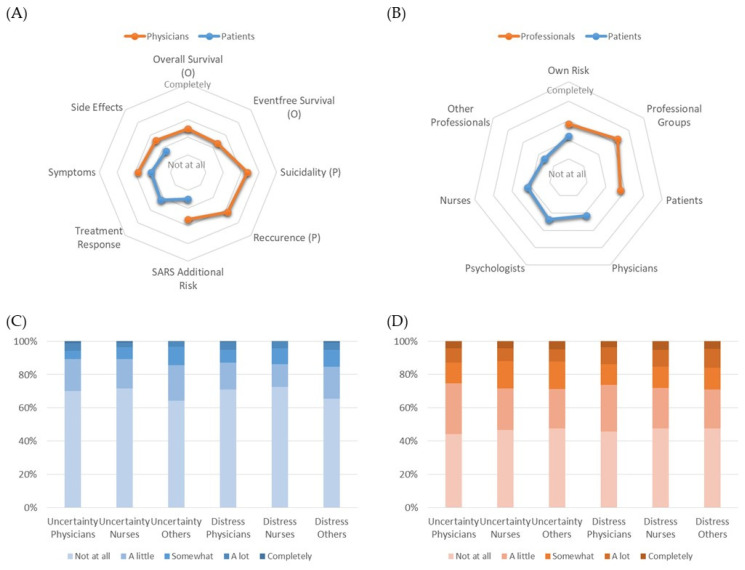
(**A**) Different perceptions of pandemic effects on criteria for treatment decisions between physicians and patients; (**B**) Perception of different aspects of burden of the opposite stakeholder group by patients and healthcare professionals; Recognition of decisional uncertainty and distress for healthcare professionals by patients in (**C**) oncology and (**D**) psychiatry. Chi² test showed significantly higher perception of burden in psychiatry compared to oncology in all six categories (*p* < 0.001).

**Table 1 cancers-14-04317-t001:** Standardized canonical discrimination function coefficients for differentiation between (A) patients in oncology and psychiatry; and (B) nurses and physicians. All included items showed significant differences of the group means (Wilks–Lambda test).

	Discrimination Function Coefficients	Wilks–Lambda	*p* Values
(A)			
Decision Support Social Environment	−0.191	0.888	0.000
Decision Support Own Evaluation	−0.145	0.947	0.000
Decision Criteria Symptoms	0.022	0.986	0.018
Burden Infection Risk	0.667	0.973	0.001
Age 3 groups	0.554	0.720	0.000
Factor Psychological Conditions	−0.754	0.755	0.000
(B)			
Burden Patients	−0.506	0.935	0.000
Fulfillment Legal Obligation	0.054	0.988	0.014
Management Hygiene	−0.358	0.941	0.000
Resources Drug Treatment	0.438	0.989	0.018
Resources Drug Availability	−0.306	0.982	0.003
Resources Diagnostics	0.341	0.976	0.008
Resources Protective Equipment	−0.142	0.976	0.001
Burden Communication	0.020	0.969	0.000
Pandemic Workload	0.414	0.944	0.000
Burden Own Risk	0.282	0.952	0.000

## Data Availability

The data that support the findings of this study are available on request from the corresponding author. The data are not publicly available due to privacy or ethical restrictions.

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
