# Peer review of "Differences in Stakeholders’ Perception of the Impact of COVID-19 on Clinical Care and Decision-Making"

_cancers, 2022, doi:10.3390/cancers14174317_

Round 1
Reviewer 1 Report
This is a very good and very sound study. The period of data collection reflects the time of the alpha variant of Covid-19. It would be interesting if the same questions were also asked to physicians, nurses and patients during delta variant transmissions. Would the figures be the same between the two periods of Covid-19 transmission.
What would also be interesting is whether the findings are similar if the analysis was to compare the results according to the type of the institutions, i.e.: university hospitals versus non-academic hospitals. Similarly, if the analysis was done based on gender sub-group, because the perceptions based on gender might provide different pictures, especially during an emergency condition such as Covid-19 pandemic.
Author Response
Thank you for the valuable comments.
We added the findings regarding gender, age groups and different types of provides in the discussion:
Interestingly, younger age (<40 years), but not gender was identified as predictor for decisional conflicts in all participating groups. Unfortunately, the structure of the questionnaire processes did not allow a differentiation between different provider types. This should be considered for future research.
We agree that the comparison of different provider types is an intersting aspect. Unfortunately, the data protection rules and the structure of recruitment did not allow the investigation of this topic.
The suggestion to continue the questionnaries for other time points is valuable and was included in the overall project. Unfortunately, the results are not ready for reporting yet. We added this to the discussion as perspective.
Further research needs to be done for evaluation of perception dynamics over the course of a pandemic and its relationship to incidence developments.
Reviewer 2 Report
This cross-sectional survey of German patients and providers evaluates the impact of COVID on care delivery. It is of potential interest but has considerable flaws.
- Unclear why this includes cancer care and psychiatry as these are quite different; would separate into entirely distinct papers.
- Need more information on pilot testing and validation of survey instrument; was this multilingual or only in German?
- No info on recruitment/sampling strategy or criteria, response rates by different categories, etc
- how was missing data managed in the analysis?
- prior paper covers decisional conflict; only focus on novel components herein
Author Response
We are thankfull for the valuable comments.
We added an explanation why psychiatry was compared with cancer care.
Previously published results from our group [5, 9] suggested that the clinical setting appears to influence the extent of decisional conflicts and impact on SDM. ...
For cancer care we hypothesized that the high prognostic, potentially life-threatening impact of diagnostic and treatment delays might be a determinant of the perception of decisional conflicts and SDM impact. In contrast, entities that are intensively related to perception disturbances can also have intensive effects on decision making for all involved groups. Therefore, cancer care was compared with psychiatry to differentiate these two aspects.
The details of the questionnaire development wer reported in our previous publication. In order to better follow the key aspects of valiudation we added the following sentence.
Validation of the questionnaire versions (available only in German language) were done in two rounds with 5 representatives in each group.
Additional data regarding the response rates were added in the results section. In order to avoid double publication we refer to previous publications regarding the recruiting strategy. This is now mentioned in the respective paragraph.
Details of recruitment were previously published [5, 9].
Handling of missing data was included in the respective paragrph.
Missing data occurred in up to 3% of the participants depending from the different items. These data were excluded from the analysis in a case-based manner.
The dedicated focus on cross-group perception of decisional conflicts and the observed dissociation were more pronounced in the discussion and by adding a conlusion section.
Patients and healthcare professionals reflect decisional conflicts that appear as result of the pandemic and related changes in clinical process management. These decisional conflicts were seen in projection of the own groups. However, this perception towards the other groups was dissociated regarding the severity of the impact and overestimated the effects of the pandemic. This dissociation can affect SDM and adapted pandemic management seems to require specific support to maintain awareness for these decisional conflicts.
Reviewer 3 Report
This is a well-written, interesting, and clinically relevant article. Research performed is also of good quality. I only have a few minor suggestions:
I would strongly recommend that the paper is reviewed and partially rewritten;
In the introduction, I suggest to do a brief description of the findings of previous studies should be include here, as well as an explanation of what the current study contributes to this literature
Author Response
We thank the reviewer for the evaluation and comments.
We added additional information about the previous results from our group in the introduction. In addition, we modified the introduction to better understand the contribution of the manuscript to the already published results.
Distinct perception profiles of changes in oncology care processes due to COVID-19 were previously identified and more than 20% of the healthcare professionals, but only ~11% of the patients reported severe decisional conflicts during the pandemic [5]. Therefore, it seems to be interesting to compare the perception of decisional conflicts and impact on SDM between different stakeholder groups.
Round 2
Reviewer 2 Report
.